# Preparation of Nanofiltration Membrane Modified with Sawdust-Derived Cellulose Nanocrystals for Removal of Nitrate from Drinking Water

**DOI:** 10.3390/membranes12070670

**Published:** 2022-06-28

**Authors:** Amos Adeniyi, Danae Gonzalez-Ortiz, Céline Pochat-Bohatier, Sandrine Mbakop, Maurice Stephen Onyango

**Affiliations:** 1Department of Chemical, Metallurgical and Materials Engineering, Tshwane University of Technology, Pretoria 0001, South Africa; sandrine.mbakop88@gmail.com; 2Water for Rural Communities (WARUC) NPC, Pretoria 0002, South Africa; 3Institut Européen des Membranes, IEM UMR-5635, Université de Montpellier, ENSCM, CNRS, Place Eugène Bataillon, CEDEX 5, 34095 Montpellier, France; danae.gonzales-ortiz@umontpellier.fr (D.G.-O.); celine.pochat@umontpellier.fr (C.P.-B.)

**Keywords:** nanofiltration, cellulose nanocrystals, nitrate, interfacial polymerization, water flux, rejection

## Abstract

In this work, cellulose nanocrystals (CNC) derived from sawdust were successfully incorporated into a nanofiltration membrane produced by the interfacial polymerization of piperazine (PIP) and trimesoyl chloride (TMC). The characteristics of unmodified and CNC-modified membranes were investigated using scanning electron microscopy (SEM), Atomic Force Microscopy (AFM), zeta potential measurement, X-ray photoelectron spectroscopy (XPS), and contact angle measurement. The performance of the membranes in terms of nitrate removal and water flux was investigated using 60 mg/L of potassium nitrate solution in a dead-end test cell. The characteristics of the modified membrane revealed a more nodular structure, higher roughness, increased negative surface charge, and higher hydrophilicity than the pristine membrane, leading to nitrate rejection of 94%. In addition, the membrane gave an average water flux of 7.2 ± 1.8 L/m^2^/h/bar. This work implies that nanofiltration, a relatively low-pressure process compared to reverse osmosis, can be used for improved nitrate removal from drinking water using an NF membrane modified with sawdust-derived cellulose nanocrystals.

## 1. Introduction

Nitrate is likely the most common groundwater contaminant on the planet, posing a significant health risk and causing eutrophication [1,2]. This is aggravated by its high water solubility. The presence of high levels of nitrate in groundwater is almost always due to pollution caused by anthropogenic activities [3]. Agriculture, which accounts for 70% of global water withdrawal, is a major contributor to water contamination [4]. Huge amounts of agrochemicals, organic matter, medication residues, sediments, and saline runoff are discharged into water bodies by farms [5,6]. In many parts of the world, nitrate concentrations in surface water, especially groundwater, have risen [7]. Specifically, in the rural areas of South Africa, where groundwater is considered the cheapest and most available drinking water source, high levels of nitrate are a major concern [8,9].

Nitrate may be successfully removed from water using treatment processes such as ion exchange and reverse osmosis. However, these are expensive treatment options. Nanofiltration provides a cheaper alternative, but a suitable membrane needs to be developed as the process has a greater capability to remove divalent ions, while nitrate is a monovalent ion. A nanofiltration membrane is considered a loose reverse osmosis membrane since its characteristic lies between ultrafiltration and tight reverse osmosis membranes [10]. The membrane can reject divalent ions but is weak at rejecting monovalent ions. However, the transmembrane pressure required is far lower than that of reverse osmosis. This makes it a process with low energy consumption and high water recovery. This is because the pore size of a nanofiltration membrane ranges from 0.5 nm to 2.0 nm [11]. Apart from this, the rejection mechanism in nanofiltration is not only size exclusion but also electrostatic interaction. This may also provide an advantage for the removal of nitrate from water if the membrane can be made to be more negatively charged.

The performance of nanofiltration membranes can be enhanced by incorporating nanomaterials [11,12]. Some of the nanomaterials that have been used include zeolite, silica (SiO_2_), titanium dioxide (TiO_2_), and silver nanoparticles (Ag NPs) [13,14]. Incorporation of these nanomaterials into the matrix of nanofiltration membranes has resulted in an increase in the flux and rejection, better mechanical and thermal stability, and anti-fouling and anti-bacterial properties. Ang et al. [15] incorporated hollow silica nanoparticles into the matrixes of nanofiltration polyamide membranes. They reported an increase in the water flux because of the presence of the nanomaterial. Soria et al. [16] reported an increase in the rejection of NaCl and increase in hydrophilicity and antimicrobial activity when polydopamine was used to firmly bind a combination of TiO_2_ and ZnO nanoparticles on nanofiltration membranes. Li et al. [17] used a tannic acid (TA) coating solution containing TiO_2_ nanoparticles as an aqueous phase for interfacial polymerization to prepare a modified polyamide nanofiltration membrane. They reported a decrease in TiO_2_ aggregation and enhanced interfacial compatibility between TiO_2_ and the polyester matrix. They also reported an increase in salt rejection and water flux. Polisetti and Ray [18] reported the improved flux and antifouling properties of a polyamide NF film prepared on nanoparticle (SiO_2_/TiO_2_)-modified polyacrylonitrile/70:30 blend ultrafiltration substrates. This is a thin-film composite with a nanocomposite substrate rather than a thin-film nanocomposite as prepared in this work.

A new type of nanomaterial that is of great interest is cellulose nanocrystals. This is because of their properties and the fact that they can be derived from waste materials such as sawdust [19,20]. Bai et al. [21] incorporated cellulose nanocrystals (CNCs) into thin-film composite (TFC) nanofiltration membranes. They reported improved desalination performance and dye removal. In another work, Bai et al. [22] reported enhanced separation performance and antifouling properties when CNC was incorporated into polyamide nanofiltration membranes. Ghaee et al. [23] fabricated a polyamide membrane using the interfacial polymerization of m-phenylenediamine (m-PDA) and 2,4-diaminobenzene sulfonic acid with trimesoyl chloride (TMC) on a polysulfone sub-layer. They reported that nitrate ion rejection was enhanced from 63% to 85%, which could be attributed to surface charge enhancement. Zou et al. [24] reported an improvement in nitrate rejection rate to 88.8% and permeate flux of 27.0 L/m^2^/h using a commercial NF membrane modified using poly(sodium 4-styrenesulfonate) (PSS). In this work, we report enhanced nitrate removal (94%) and water permeability (7.2 L/m^2^/h/bar) by the incorporation of sawdust-derived cellulose nanocrystals into polyamide thin-film nanofiltration membranes. The aqueous and organic phase concentration were prepared based on the result of Tian et al. [25]. CNC at 0.21 wt %, obtained after several trials in our previous work, was added to the aqueous solution [26].

## 2. Materials and Methods

### 2.1. Materials

Piperazine (PIP), trimesoyl chloride (TMC), hexane, sodium dodecyl sulfate, and potassium nitrate were used for the study. They were obtained from Sigma-Aldrich, Centurion, South Africa, and were analytical-grade chemicals. The cellulose nanocrystals (CNC) and polysulfone UF membrane (100 kDa) used were obtained from CSIR Durban, South Africa and Microdyn Nadir, Johannesburg, South Africa respectively. Deionized water was produced using the Purite water system (Model Select Analyst HP40, United Kingdom). The quality was between 14 and 15 mΩ.

### 2.2. Membrane Synthesis

Two membranes were synthesized using the interfacial polymerization method and named PT (unmodified) and PTUC (modified). The monomers were 6 g piperazine (PIP) in 100 mL water (aqueous phase) and 0.3 g trimesoyl chloride (TMC) in 100 L hexane (organic phase). The UF polysulfone membrane was soaked in a solution containing 0.5 percent sodium dodecyl sulfate (SDS) for 12 h. The membrane was then rinsed and soaked in deionized water for 2 h before being dried in a fume cupboard for 1 hour. The membrane was then adhered to a glass plate using double-sided tape. In the preparation of PT, the aqueous solution was poured on the UF membrane and allowed to stay for 30 min. The excess water solution was wiped away from the membrane’s surface with filter papers, and the organic solution was then applied. For 60 s, the reaction was allowed to run. The fabricated membrane was allowed to drain before being treated in a 65 °C oven for 15 min. The membrane was rinsed and immersed in deionized water after it had been made. In the case of PTUC membrane preparation, the same procedure as for the preparation of the PT membrane was used except that 0.21 g of cellulose nanocrystals (CNC) was added to the aqueous solution based on our previous work [26].

### 2.3. Membrane Characterization

The membranes were characterized for elemental composition using X-ray photoelectron spectroscopy (XPS), for the surface charge using zeta potential measurement, for morphology using scanning electron microscopy (SEM) and atomic force microscopy (AFM), and for hydrophilicity using contact angle measurement. The procedures for SEM, AFM, and contact angle measurement have been described in our previous work [26]. An X-ray photoelectron spectrometer (Thermo, Model ESCAlab 250Xi, Waltham, MA, USA) with Monochromatic Al kα (1486.7 eV) was used to study the chemical compositions of their polyamide layers.

#### Zeta Potential Measurement

The measuring device was a Surpass electrokinetic analyzer (Anton Paar, Graz, Austria) equipped with an adjustable “gap” measuring cell. The measuring surface was 20 mm × 10 mm. The fibers were placed on two sample holders using a double-sided adhesive and the gap between the two membrane sample surfaces was 100 µm. Three different solutions were prepared for the analysis: electrolyte solution (KCl 10-3 M at pH 11), acid solution (HCl 10-1M), and basic solution (NaOH 10-1M). The traffic flow of the electrolyte solution (10-3 M of KCl) was 0 to 500 ± 0.5 mL/min, with a pressure maximum differential of 300 mbar. All measurements were carried out with a flow of approximately 100 mL/min and a maximum pressure of 300 mbar. To vary the pH in the beaker containing the electrolyte solution, automatic titration was provided by injection of micro-volumes of acid (HCl) solution. Thus, the range of pH was from the initial pH of KCl around 11 to the acid pH. Before starting any measurement, the symmetry of the flow channel was verified by monitoring the evolution of the pressure as a function of the circulation flow in both directions of flow. The procedure was then validated by checking a flow perfectly proportional to the pressure (linearity of the flow).

### 2.4. Water Treatment Performance Tests

The performance tests in terms of water flux and nitrate (60 mg/L) rejection were done using a dead-end test cell. The active membrane area of the test cell was 14.6 cm^2^. A concentration of 60 mg/L was chosen because the acceptable limit for drinking water is 50 mg/L according to WHO guidelines [27]. The membranes were first compacted at a pressure of 5 bar for 8 h. The pure water flux was investigated at a pressure ranging from 5 bar to 10 bar. Permeate flux and rejection for nitrate solution were investigated at a pressure ranging from 9 bar to 13 bar. The conductivity of the feed and final permeate was determined using a DR/890 Colorimeter. Equations (1) and (2) were used to calculate the nitrate rejection and the permeate flux, respectively.
(1)$$J_{w}=\frac{V}{At}$$
(2)$$R=\frac{\left(C_{f}-C_{p}\right)}{C_{f}}*100$$
where $J_{w}$ (L/m^2^/h) is the water flux, V (L) is the permeate volume, *R* is the rejection in %, *C_f_* is the solute concentration in the feed, and *C_p_* is the solute concentration in the permeate. The filtration time t is in h and the active membrane area *A* is measured in m^2^.

## 3. Results and Discussion

### 3.1. Membrane Characteristics

#### 3.1.1. Morphology and Surface Structure

Figure 1 shows the change in membrane morphology as a result of adding 0.21% CNC into the aqueous phase used for the polymerization reaction.

Both membranes had a nodular structure and a rough surface. This confirms that the film was made on a polysulfone blend support. All of the membranes had a nodular structure, showing that polyamide membranes were fabricated via interfacial polymerization. The sizes of nodules and nodule aggregates are critical when considering membrane surface characteristics. A nodule is a clump of polymer molecule agglomerates that are always located near the surfaces of polymeric membranes and are entangled with one another. Membrane “peaks” and “valleys” are caused by nodular formations, which make the membrane rougher. Due to the abundance of water absorption sites, rough membranes are known to have higher water permeability [28]. The nodular structure was observed to be more pronounced and larger when 0.21% CNC was added to the membrane formulation. This led to an increase in the RMS roughness from 10.47 to 77.99 nm.

#### 3.1.2. Surface Charge

Figure 2 contains the results of the zeta potential measurement, showing a significant change in the surface charge of the membrane as a result of the presence of cellulose nanocrystals.

Cellulose nanocrystals are negatively charged [29] and can impact the surface charge of a membrane if a successful modification has taken place [12]. Membrane surface charge characteristics are thought to be important in determining salt rejection and permeate flux decline. Membranes electrostatically reject charged organic molecules, colloids, and particles due to their charged surfaces at neutral pH values, which ultimately plays a vital role in foulant accumulation on membrane surfaces, which is linked to a decrease in permeate flux [30]. At high pH, the negative zeta potential of both membranes diminished, and the isoelectric point migrated towards the acidic range, which is evidence that both membranes are hydrophilic [31]. The zeta potential of the modified membrane, on the other hand, was lower than that of the unmodified. This indicates that the membrane has been successfully modified. This is likely to increase solute rejection, particularly negatively charged nitrate, and reduce the fouling tendency [32].

#### 3.1.3. Atomic Concentration

The atomic concentration of the membrane was explored using an XPS probe of the surface of the top polyamide layer. The support polysulfone layer was not probed. As expected for polyamide membranes, the scan revealed predominantly the presence of C, O, and N. The composition is shown in Table 1 and Figure 3. An improvement in the N/O ratio was observed in PTUC. This may be due to the presence of cellulose nanocrystals.

For PT, the intensity of the O (1s), N (1s), and C (1s) peaks was at 531.9, 339.9, and 284.9 eV, respectively, while for PTUC, the intensity of the O (1s), N (1s), and C (1s) peaks was at 531.4, 339.8, and 284.9 eV, respectively. This confirmed the formation of the polyamide membrane in both cases [33].

The spectra of the two membranes were composed of one major (weak electron-withdrawing environment) and one minor peak (strong electron-withdrawing environment). For PT, the major and minor peaks were centered at 284.6 and 288.7 eV, respectively. The major peaks included three peaks underneath. The first peak at 284.6 eV was assigned to C-C and C-H aliphatic and aromatic bonds. The second one (284.2) was assigned to C-C and the third (285.8) was associated with carbons in a relatively higher electron-withdrawing environment (C-O and C-N). For PTUC, the major and minor peaks were centered at 284.8 and 288.7 eV, respectively. The major peaks included two peaks underneath. The first peak at 284.8 eV was assigned to C-C and C-H aliphatic and aromatic bonds. The second one was associated with carbons in a relatively higher electron-withdrawing environment (C-O and C-N). For both membranes, the minor peaks contained two peaks after deconvolution. The first peak was assigned to carbonyl groups and the second could be assigned to a higher electron-withdrawing environment caused by esters. Since both membranes were piperazine-based, the binding shift was smaller due to less electron sharing by the piperazine monomer. The multi-region spectra of C 1s, O 1s, and N1s for both membranes are shown in Figure 3, which indicate the presence of aromatic/aliphatic and amine atoms typical of nanofiltration membranes [34].

#### 3.1.4. Hydrophilicity

The hydrophilicity of the membranes is reflected in terms of the contact angles of water dropped on the membrane surface. Results are shown in Figure 4. The modified membrane PTUC was observed to be more hydrophilic, with a lower contact angle (the lower the contact angle, the higher the hydrophilicity). The contact angles for PTUC and PT are 70.6 ± 0.2° and 81.4 ± 0.1°, respectively. This can be attributed to CNC, because of the additional hydroxyl group from the CNC. The surface chemistry of the membrane might have been affected by the high water affinity of the cellulose nanocrystals. The pore surface oxygen-containing group has been modified by CNC, affecting the interaction between the orientation of water at the solid–liquid interface and the hydrophilicity of the solid surface. The capability of the water molecule to form hydrogen bonds created a stronger interaction between the solid phase and water in the polymer matrix, because of the incorporation of the CNC [35].

### 3.2. Comparing the Performance of the Membranes

Figure 5A shows the membranes’ performance in terms of pure water flux against the applied pressure. The pure water flux increases with pressure, as expected. It was observed that the water flux was higher for PTMUC than PT. This observation is likely due to many reasons that include higher surface roughness, higher hydrophilicity, and higher negative charge. All of these reasons are due to the effect of modifying the nanofiltration membrane (PT) with cellulose nanocrystals. This may be because of an increase in the hydrophilicity of the modified membrane due to the addition of the OH group to the polyamide film. Figure 5B shows the water flux when 60 mg/L potassium nitrate solution was utilized, while Figure 5C shows the nitrate rejection.

The pattern for the water flux when a solution of 60 mg/L of potassium nitrate was utilized with both membranes is similar to the pure water flux. The exemption is that the water flux was reduced and was obtained at higher operating pressures. This is due to the osmotic pressure that has to be overcome by the applied pressure to obtain water flux across the membrane. At all pressures investigated, the water flux was higher with the modified membrane PTUC. Average water flux for PTUC was 7.2 ± 1.8 L/m^2^/h/bar, and for PT, it was 4.2 ± 1.0 L/m^2^/h/bar. Average nitrate rejection for both PTUC and PT membranes was found to be 94 ± 1.97 and 91 ± 3.1%, respectively. The rejection is high for both membranes primarily because of the constituent of both the aqueous and organic phases of the monomers [25] and also the surface charges of the membranes [24]. The surface zeta potential for the modified membrane was much higher than that of the pristine membrane, implying that the increased negative charge density strengthened the electrostatic interaction between nitrate and membrane, thus enhancing the nanofiltration membrane’s rejection ability. The rejection of nitrate for the PMUC might have been higher due to an increase in negative charge density, higher roughness, and hydrophilicity [23,36]. However, the separation mechanism for the rejection of nitrate is dual, which means both that steric exclusion and Donnan exclusion are involved [37].

## 4. Conclusions

In this work, a nanofiltration membrane modified by the incorporation of sawdust-derived cellulose nanocrystals (CNC) was fabricated for enhanced nitrate removal using the interfacial polymerization method. The organic phase contained TMC as the monomer, while the aqueous phase containing the CNC had piperazine as a monomer. The CNC-modified membrane was compared with the unmodified one in terms of characteristics and performance. The characteristics of each membrane were examined using SEM, AFM, zeta potential measurement, XPS, and contact angle measurement. The modified membrane had a more nodular structure, higher roughness, increased negative surface charge, and higher hydrophilicity. Consequently, nitrate rejection of 94% was obtained. In addition, the membrane gave an average water flux of 7.2 ± 1.8 L/m^2^/h/bar. This means that nanofiltration, a relatively low-pressure process compared to reverse osmosis, can be used for improved nitrate removal from drinking water using an NF membrane modified with sawdust-derived cellulose nanocrystals.

## Figures and Tables

**Figure 1 membranes-12-00670-f001:**
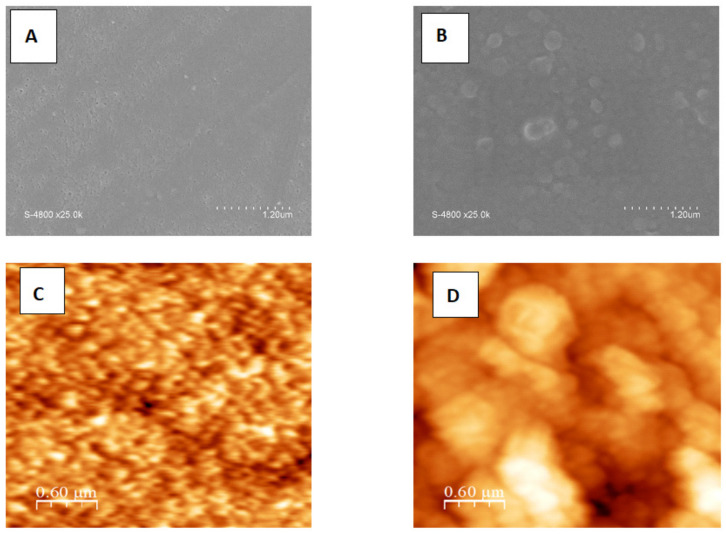
Morphology of membrane surfaces as revealed by SEM and AFM. (**A**,**B**) show the SEM surface images of PT and PTUC. (**C**,**D**) show 2D AFM images of the surface of PT and PTUC. (**E**,**F**) show 3D AFM images of the surface of PT and PTUC. (**G**,**H**) show AFM surface roughness of the surface of PT and PTUC.

**Figure 2 membranes-12-00670-f002:**
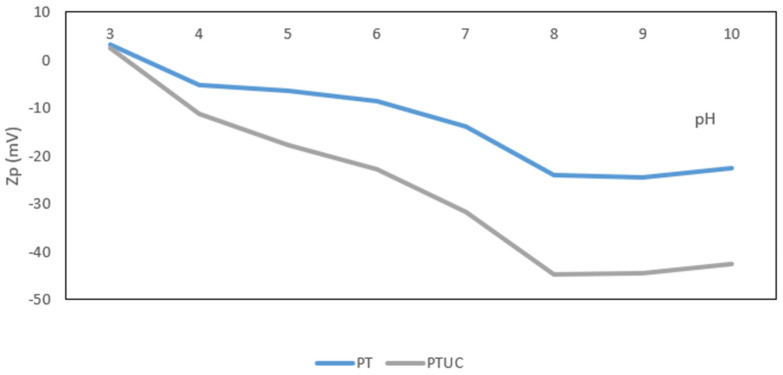
pH dependence of zeta potential for unmodified membrane (PT) and the membrane modified with sawdust-derived cellulose nanocrystals (PTUC). The figure shows a change in the surface charge as a result of the modification.

**Figure 3 membranes-12-00670-f003:**
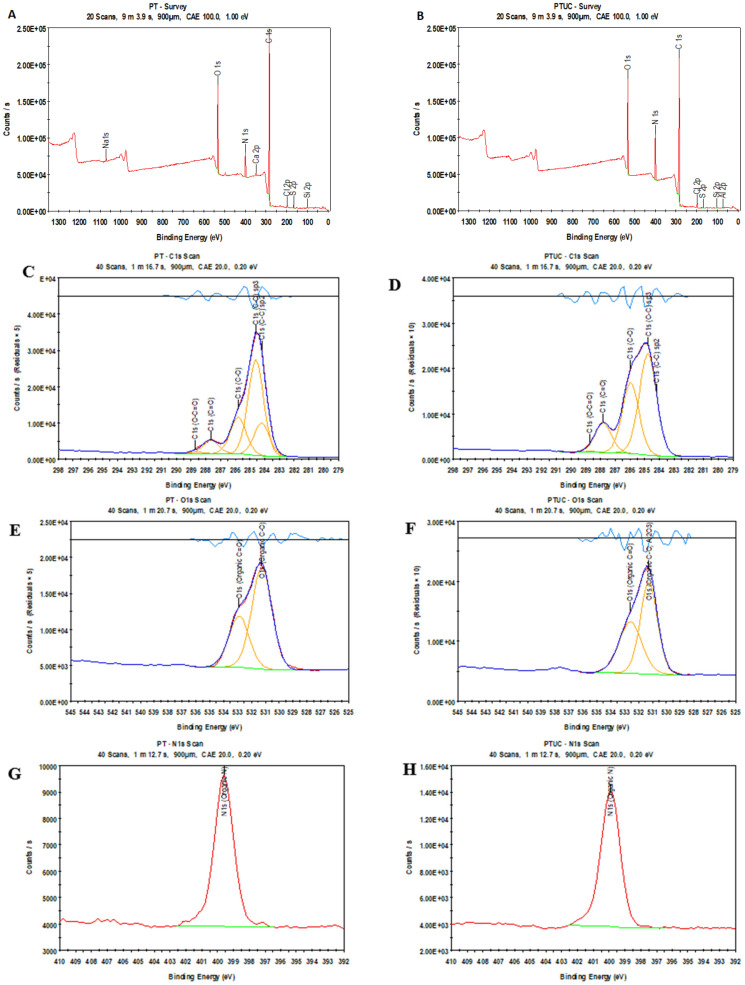
Survey spectrum of PT (**A**) and PTUC (**B**). C 1s, O 1s, and N 1s core-level spectra of PT (**C**,**E**,**G**) and PTUC (**D**,**F**,**H**).

**Figure 4 membranes-12-00670-f004:**
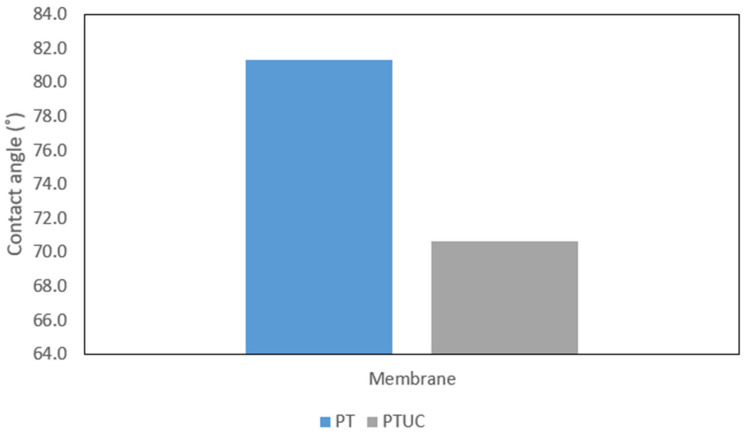
Comparing contact angles for the membranes.

**Figure 5 membranes-12-00670-f005:**
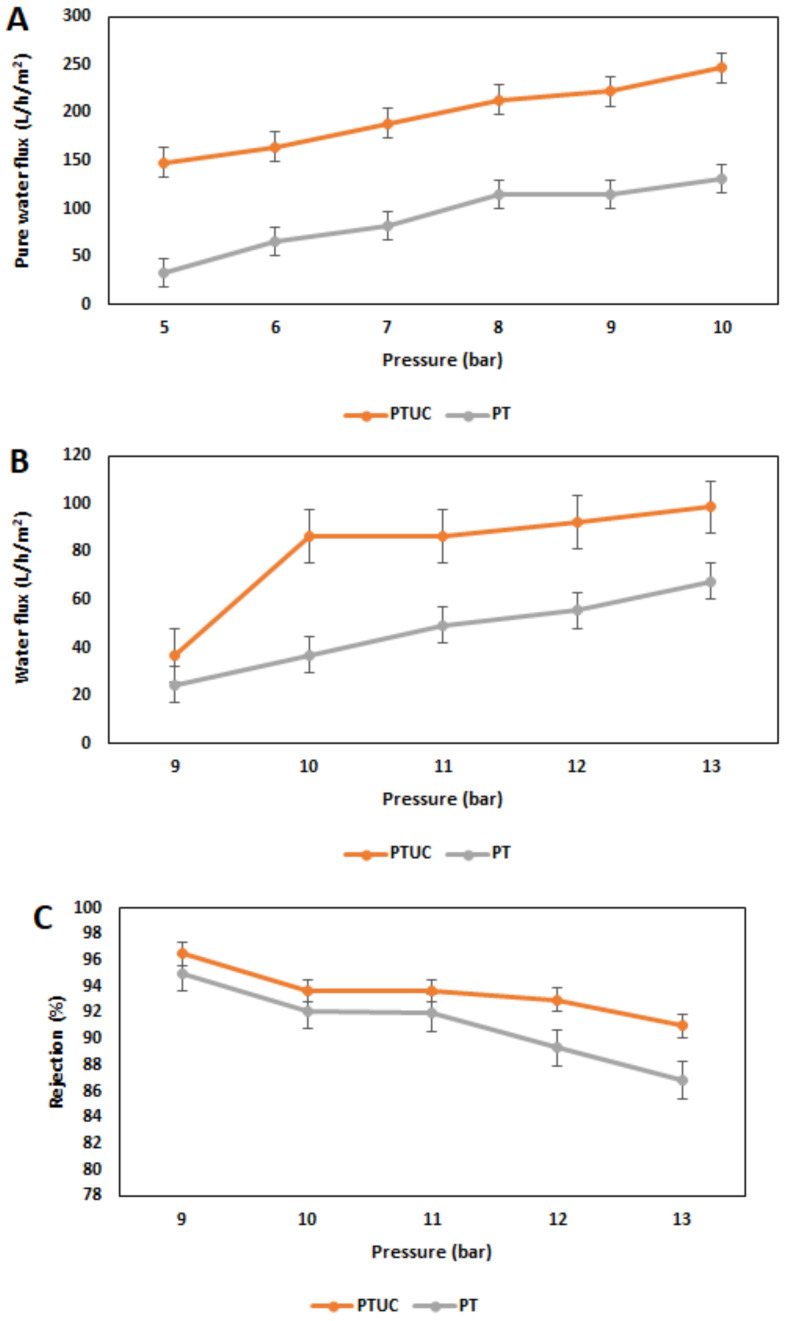
(**A**) Pure water flux. (**B**) Water flux for a solution containing 60 mg/L potassium nitrate. (**C**) Nitrate rejection.

**Table 1 membranes-12-00670-t001:** Elemental compositions of the membranes.

Membrane	Elemental Composition			O/C	N/C	N/O
	C	O	N			
PT	76.00	16.1	5.9	0.21	0.08	0.37
PTUC	73.06	14.92	12.02	0.20	0.16	0.81

## Data Availability

Not applicable.

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
