# Peer review of "Preparation of Nanofiltration Membrane Modified with Sawdust-Derived Cellulose Nanocrystals for Removal of Nitrate from Drinking Water"

_membranes, 2022, doi:10.3390/membranes12070670_

Round 1

Reviewer 1 Report

The manuscript by Amos Adeniyi. et al. studied the influence of sawdust derived cellulose nanocrystals on the flux and antifouling properties of nanofiltration membrane. The idea sounds interesting; however, sawdust derived cellulose nanocrystals are not ideal for typical nanofiltration applications. Besides, the experiments are poorly done, and the figures are not organized and described correctly. Here are some concerns please follow:

1. In figure 1, the AFM images are not aligned. There aren't any scale bars or units for these figures both the x, y and z-axis.

2. In line 58, silica (SiO2), titanium dioxide (TiO2), and silver nanoparticles (Ag NPs) change to scientific units like SiO2 and TiO2. And also, the citation given is very general, the author may need to cite the latest literature here are some on SiO2 and TiO2 based UF and polyamide nanofiltration membranes. Journal of Applied Polymer Science 138.16 (2021): 50228. (https://doi.org/10.1002/app.50228) and Journal of Applied Polymer Science 138.1 (2021): 49606. (https://doi.org/10.1002/app.49606)

3.    In figure 1 G and F Roughness images are not clear It looks blurry Please use the high-resolution image. There are no units for contact angle in Figure 4.

4.    If possible, please provide a characterization of the cellulose nanocrystals to understand how the CNC is embedded in the polyamide layer like morphological images etc.

5.    What is the separation mechanism of nitrate is it steric(sieving) or electrical (Donnan) effect? Explain? 

Author Response

  1. In figure 1, the AFM images are not aligned. There aren't any scale bars or units for these figures both the x, y and z-axis. Done
  2. In line 58, silica (SiO2), titanium dioxide (TiO2), and silver nanoparticles (Ag NPs) change to scientific units like SiO2and TiO2. And also, the citation given is very general, the author may need to cite the latest literature here are some on SiO2 and TiObased UF and polyamide nanofiltration membranes. Journal of Applied Polymer Science 138.16 (2021): 50228. (https://doi.org/10.1002/app.50228) and Journal of Applied Polymer Science 138.1 (2021): 49606. (https://doi.org/10.1002/app.49606)

The first reference (https://doi.org/10.1002/app.50228) has been included, but the other is about ultrafiltration membrane and cannot be added

  1.   In figure 1 G and F Roughness images are not clear It looks blurry Please use the high-resolution image. There are no units for contact angle in Figure 4. Done
  2.   If possible, please provide a characterization of the cellulose nanocrystals to understand how the CNC is embedded in the polyamide layer like morphological images etc. I do not have this image

  1.   What is the separation mechanism of nitrate is it steric(sieving) or electrical (Donnan) effect? Explain? Done

Reviewer 2 Report

The authors describe the preparation of NF membranes impregnated with cellulose nanocrystals for nitrate rejection. Even though, the nitrate removal is a very important field in the ground water remediation, there are several more efficient technologies that are already available. RO can perform the same task without the additional energy input for an NF pre-treatment. So the significance of this work, in terms of advantages offered in comparison to existing methods is not well established. In addition, the actual validation of the objective is very minimal with just one composition (0.21%) CNC) evaluated with a single salt solution. In reality, the groundwater is much more complex and has many other constituents. Even if it is difficult to test the actual groundwater, at least a mixture of salts including Na+, Ca2+, Mg2+, Cl-, etc should be tested and the selectivity factor must be determined. Also, is there any reason why only one composition of CNC was tried? The nitrate rejection rate is not much higher than the pristine membrane but only beneficial in terms of flux. There is no way to tell if this is an optimized formulation. I recommend that the manuscript be revised to include the tests suggested above and also address the following minor issues before reconsideration. 

1. Most of the manuscript is not in the standard journal template. Please revise to make sure to conform with the template, including indenting, citations, etc. 

2. Abstract." higher nodular structure, higher..." Please specify what are these proper higher in comparison to?

3. Line 58, "2" in molecular formulas (SiO2, TiO2) must be sub-scripted.

4. Line 84, water flux (7.2...) must be changed to water permeability.

5. Line 87, 0.21 w%....weight percent is usually reported as wt.%

6. Line 146, acceptable limit of 50 mg/L..Please specify if this is the acceptable limit for drinking water applications?

7. Lines 161, 162, formatting error.

8. Figure 3 should be in the Section 3.1.3.

9. Figure 4. Please include the standard deviation values.

10. Figure 5(b) y-axis. 2 in "m2" should be super-scripted.

11. References must be reformatted according to the journal template. 

Author Response

0.21 wt % was obtained after several trial in our previous work and we established the fact that we are using this in the work in the introduction. Getting a good membrane depends on the homogeneity of the aqueous solution which was obtained at 0.21 wt %.

The plan was to show the potential for nitrate removal and did not include selectivity in which case testing with raw water would have been the most appropriate.

  1. Most of the manuscript is not in the standard journal template. Please revise to make sure to conform with the template, including indenting, citations, etc. done
  2. Abstract." higher nodular structure, higher..." Please specify what are these proper higher in comparison to? done
  3. Line 58, "2" in molecular formulas (SiO2, TiO2) must be sub-scripted. done
  4. Line 84, water flux (7.2...) must be changed to water permeability. done
  5. Line 87, 0.21 w%....weight percent is usually reported as wt.% done
  6. Line 146, acceptable limit of 50 mg/L..Please specify if this is the acceptable limit for drinking water applications? done
  7. Lines 161, 162, formatting error. done
  8. Figure 3 should be in the Section 3.1.3. done
  9. Figure 4. Please include the standard deviation values. (included in the text but not on the figure)
  10. Figure 5(b) y-axis. 2 in "m2" should be super-scripted. done
  11. References must be reformatted according to the journal template. done

Round 2

Reviewer 2 Report

The authors have addressed most of the issues raised.